# Role of Toothbrushes as Gene Expression Profiling Tool for Oral Cancer Screening in Tobacco and Alcohol Users

**DOI:** 10.3390/ijerph19138052

**Published:** 2022-06-30

**Authors:** Govindarajan Sujatha, Vishnu Priya Veeraraghavan, Ahmed Alamoudi, Maha A. Bahammam, Sarah Ahmed Bahammam, Yaser Ali Alhazmi, Hazar S. Alharbi, Khalid J. Alzahrani, Mohammad S. Al-Ghamdi, Fuad M. Alzahrani, Saranya Varadarajan, A. Thirumal Raj, Shankargouda Patil

**Affiliations:** 1Department of Oral Pathology and Microbiology, Sri Venkateswara Dental College and Hospital, Chennai 600130, India; gsuja@rediffmail.com (G.S.); vsaranya87@gmail.com (S.V.); thirumalraj666@gmail.com (A.T.R.); 2Saveetha Institute of Medical and Technical Sciences, Saveetha Dental College and Hospitals, Saveetha University, Chennai 600130, India; 3Centre of Molecular Medicine and Diagnostics (COMManD), Department of Biochemistry, Saveetha Dental College & Hospitals, Saveetha Institute of Medical and Technical Sciences, Saveetha University, Chennai 600130, India; drvishnupriyav@gmail.com; 4Oral Biology Department, Faculty of Dentistry, King Abdulaziz University, Jeddah 21589, Saudi Arabia; ahmalamoudi@kau.edu.sa; 5Department of Periodontology, Faculty of Dentistry, King Abdulaziz University, Jeddah 21589, Saudi Arabia; mbahammam@kau.edu.sa; 6Executive Presidency of Academic Affairs, Saudi Commission for Health Specialties, Riyadh 11614, Saudi Arabia; 7Department of Pediatric Dentistry and Orthodontics, College of Dentistry, Taibah University, Medina 42353, Saudi Arabia; sbahammam@taibahu.edu.sa; 8Department of Oral and Maxillofacial Surgery and Diagnostic Sciences, College of Dentistry, Jazan University, Jazan 45412, Saudi Arabia; dr.y.alhazmi@gmail.com; 9Department of Basic Dental Sciences, College of Dentistry, Princess Nourah Bint Abdulrahman University, Riyadh 11671, Saudi Arabia; hasalharbi@pnu.edu.sa; 10Department of Clinical Laboratories Sciences, College of Applied Medical Sciences, Taif University, Taif 21944, Saudi Arabia; ak.jamaan@tu.edu.sa (K.J.A.); abo_fares_@hotmail.com (M.S.A.-G.); fuadmubarak@tu.edu.sa (F.M.A.); 11Department of Maxillofacial Surgery and Diagnostic Sciences, Division of Oral Pathology, College of Dentistry, Jazan University, Jazan 45412, Saudi Arabia; 12Centre of Molecular Medicine and Diagnostics (COMManD), Saveetha Dental College & Hospitals, Saveetha Institute of Medical and Technical Sciences, Saveetha University, Chennai 600130, India

**Keywords:** alcohol, oral cancer, RNA, toothbrush, tobacco

## Abstract

Aim: The use of toothbrushes was investigated as a potential RNA source and gene expression profiling tool for oral cancer screening in tobacco and alcohol users. Methodology: A total of 20 subjects were selected on the basis of inclusion and exclusion criteria. They were divided into two groups: group I—healthy controls (*n* = 6); group II—individuals who consume tobacco and alcohol (*n* = 14). After the volunteers brushed their teeth using a soft-bristle toothbrush with ~0.5 gm of toothpaste, the toothbrushes were collected, and the gene expression of BAX, BCL2, CDK4, CKDN2A, GNB3, and TCF7L2 was assessed. Results: The gene expression of BAX decreased significantly in alcoholics and smokers (0.13867 ± 0.12014), while the gene expression of BCL2 increased in alcoholics and smokers (1.91001 ± 0.90425) in comparison with healthy controls (*p* = 0.0054 and *p* = 0.0055). Although there was increased expression of CDK4, CKDN2A, and TCF7L2 and decreased expression of GNB3 in smokers and alcoholics, the results were not significant. Conclusions: A toothbrush is a good source of RNA, and gene expression analysis can be performed using the genetic material retrieved from toothbrushes, which can aid in the early diagnosis of oral squamous cell carcinoma among tobacco and alcohol users. Further studies with a larger sample size and different durations of toothbrush use should be conducted to explore the role of toothbrushes as a noninvasive tool for disease diagnosis.

## 1. Introduction

Oral cancer is a major cause of death in developing countries, with 90% of death seemingly due to oral squamous cell carcinoma (OSCC) [1]. The disease is multifactorial in etiology with a complex pathogenesis, including prolonged exposure to a carcinogen. Although several etiologic factors are known to cause the disease, tobacco and alcohol account for 40% of cancers and 60% of fatal cases [2]. Nicotine, the major component of tobacco, is responsible for genetic and epigenetic alterations of cancer-related genes. The harmful compounds in tobacco cause DNA damage, thereby reducing its ability to be repaired during replication, which may cause diseases related to the respiratory system [3,4]. Studies have also reported the consequences of long-term exposure to nicotine, which usually causes attention deficits and mood disorders [5] A study also reported that nicotine exposure in any age group can reduce insulin sensitivity and contribute to diabetes mellitus [6]. Prolonged usage of nicotine contributes to the risk of developing cardiovascular diseases, as well as affects the immune system. Among females, exposure to nicotine during pregnancy can impact fetal brain development and may lead to preterm delivery and low birthweight [7]. Studies have also shown that alcohol consumption is associated with smoking. In other words smokers are more vulnerable to becoming addicted to alcohol consumption [8]. Smoking, along with consumption of alcohol, leads to a higher chance of developing oral cancer [9,10,11,12,13,14].

Considering the pathogenesis following prolonged exposure of the oral mucosa to these carcinogens, several complex genetic and molecular mechanisms occur that lead to increased mitosis, evasion of apoptosis, and immune surveillance. Despite several recent advances in the management of the disease, the prognosis and 5 year survival rate have not seen much improvement. Hence, early diagnosis and treatment are vital to improve the prognosis of the disease. Several noninvasive tools are being explored to aid in early diagnosis and disease prevention. The most common genes altered in potentially malignant diseases, cancer, and tobacco users include apoptosis-related genes BAX and BCL2 and cell-cycle-related genes such as cyclin-dependent kinases CDKN2A and CDK4 [15,16]. In addition, the gene expression of *GNB3* and TCF7L2 has been linked to alcohol and nicotine addiction, respectively [17,18].

Saliva is commonly used as a noninvasive tool for the diagnosis of various diseases including oral cancer [19]. However, it is sensitive to the technique used, while the collection and storage of samples are cumbersome. In this regard, toothbrushes have been recently explored as an accurate tool in genetic analysis for person and gender identification in forensic dentistry. However, the expression of apoptosis- and cell-cycle-related genes in the genetic material retrieved from used toothbrushes has not been assessed. Therefore, this study aimed to evaluate the use of toothbrushes as a potential RNA source and gene expression profiling tool for oral cancer screening in tobacco and alcohol users.

## 2. Materials and Methods

The Institutional Ethics Committee (Ref No: IHEC/SDC/PhD/O PATH-1759/18/398) approved the study.

### 2.1. Sample Collection and Preparation

A total of 20 subjects (four females and 16 males; age: 26–52 years) were selected on the basis of the inclusion and exclusion criteria. They were divided into two groups: group I—healthy controls (*n* = 6); group II—individuals who consume tobacco and alcohol (*n* = 14). The volunteers were asked to brush their teeth using a soft-bristle toothbrush with ~0.5 g of toothpaste. Colgate^®^ toothpaste was used for all subjects.

### 2.2. Inclusion Criteria

Group I included systemically healthy patients with an absence of harmful habits. Group II included individuals with a cigarette smoking habit, along with consumption of alcohol (higher risk of oral cancer), i.e., individuals with both a current smoking habit and a previous history of smoking a minimum of 100 cigarettes in the last 5 years [20].

### 2.3. Exclusion Criteria

Group I excluded individuals characterized by other systemic diseases, complications, habits such as the use of tobacco and alcohol, pregnancy, and lactation. Group II excluded individuals characterized by other systemic diseases, complications, pregnancy, and lactation.

### 2.4. Sample Collection

After brushing, the toothbrushes were washed gently and the bristles were dipped in phosphate-buffered saline (PBS) (Sigma, St. Louis, MO, USA). Later, the brushes were stored at 4 °C until transfer to the laboratory.

The bristles were cut using a sterile blade and rinsed thoroughly in PBS. Subsequently, the floating bristles were removed and discarded. The sample was centrifuged for 10 min at 3000 RPM. The supernatant was discarded, and the pellets at the bottom were further examined for total RNA isolation.

### 2.5. Real-Time Quantitative Polymerase Chain Reaction (RT-qPCR) for Quantitative Analysis of Gene Expression

RT-qPCR was performed as described previously [16,21]. The RNA extraction from the pellets was performed using a Gene JET RNA purification kit (Thermo Scientific, Vilnius, Lithuania). Multiskan SkyHigh (Thermo Scientific, Waltham, MA, USA) was used for RNA quantification. RNA (2 μg) was reverse-transcribed using a cDNA synthesis kit (High Capacity, Applied Biosystems, Carlsbad, CA, USA) as per the guidelines of the manufacturer. Then, 1.8 μL of cDNA was used for the total reaction volume including 20 μg for each gene. Later, 5 μL of SYBR Green PCR master mix (Applied Biosystems, Austin, TX, USA) (total reaction volume of 10 μL) was used for quantitative analysis of the genes of interest on a real-time PCR system (Quant Studio 5, Applied Biosystems, Foster City, CA, USA). The expression of target genes was normalized to ACTB as the reference gene using the ΔΔCt method. Data quantification was performed using the 2^−ΔΔCt^ technique.

### 2.6. Genes Assessed

Apoptosis-related gene expression: The gene expression of BAX and BCL2 was assessed in the isolated RNA. The ratio of BAX to BCL2 was calculated to determine the overall effect on the apoptotic status.

Cell proliferation-related gene expression: The gene expression of cyclin-dependent kinases CDKN2A and CDK4, as well as their respective inhibitors, was assessed to determine the proliferation status.

### 2.7. Expression of Metabolic-Related Genes

The gene expression of *GNB3* and TCF7L2 was assessed using the primers depicted in Table 1. ACTB encoding beta (β)-actin was used as the housekeeping gene. The list of primers (Eurofins) and genes is given in Table 1. PCR cycling conditions are summarized in Table 2.

### 2.8. Statistical Analysis

Statistical analysis was performed using GraphPad Prism 8 software (GraphPad Software, La Jolla, CA, USA), with a statistical significance of *p* < 0.05 (* *p* < 0.05 and ** *p* < 0.01). The results were tabulated as the mean ± standard deviation of three replicates for each individual experiment. All experimental groups were compared with each other using an unpaired *t*-test (two-tailed).

## 3. Results

There was a significant decline in the gene expression of BAX in alcohol and tobacco users (0.13867 ± 0.12014) in comparison with healthy controls (0.73645 ± 0.53477) (*p* = 0.0054 **) and a significant increase in the gene expression of BCL2 in alcohol and tobacco users (1.91001 ± 0.90425) in comparison with healthy controls (0.84895 ± 0.36261) (*p* = 0.0055 **). The gene expression of CDK4 (*p* = 0.5144) and CDKN2A (*p* = 0.2901) was comparatively high in tobacco and alcohol users, but the values were not statistically significant. The gene expression of GNB3 (*p* = 0.6767) and TCF7L2 (*p* = 0.3832) showed no statistically significant difference in the two groups (Table 3). Results are summarized in Figure 1.

## 4. Discussion

The use of tobacco and alcohol among individuals has increased over the years, leading to a high global incidence of head and neck cancers. Hence, early detection can aid in the diagnosis and early prevention and management of the condition, thereby improving the prognosis. Several studies have shown microscopic alterations in tobacco and alcohol users via exfoliative cytology, although biopsy is the gold standard. Saliva has also been explored as a diagnostic tool for oral cancer detection. However, toothbrushes have not yet been used for detecting gene expression among tobacco and alcohol users. Therefore, this study explored the potential application of used toothbrushes as a gene expression profiling tool for oral cancer screening in tobacco and alcohol users. Since tobacco users predominantly consume alcohol, together responsible for 80% of oral cancer in men, the present study included both tobacco and alcohol users [12,13,14].

The gene expression of BAX was significantly increased in smokers and alcohol users compared with healthy controls. It is a well-known fact that the BAX gene (Bcl-2-associated X-protein) is a proapoptotic gene of Bcl-2 family, encoding the BAX-alpha protein, which plays a vital role in the intrinsic pathway of apoptosis [22,23]. The decreased expression of BAX in group II was attributed to the fact that tobacco and alcohol are associated with carcinogenesis, which is marked by apoptosis evasion. The gene expression of BCL2 was high in tobacco and alcohol users when compared with healthy controls. These findings are in accordance with the results of Tekna et al. and Rahmani et al. who presented the aberrant expression of BCL2 in tobacco users and concluded that BCL2 expression could play a direct role in tumorigenesis through apoptosis evasion [24,25].

Considering the proliferation markers involved in the cell cycle, CDK4 and CDKN2A, although the results were not significant, the expression of both genes was higher in tobacco and alcohol users when compared with healthy controls. This might be related to the sample size in the present study. However, a higher expression of these genes indicates future malignant transformation, as studies have shown the association between increased expression of cyclins and cyclin-dependent kinases in potentially malignant disorders and oral squamous cell carcinoma. *CDKN2A* has been shown to be involved in tumorigenesis via regulation of mitosis, cell death, and cell-cycle progression in the G1/S phase [26,27]. Zhou et al., in their systematic review, reported significant increased methylation of the *CDKN2A* gene in head and neck carcinogenesis with an odds ratios (ORs) of 6.72, *p* < 0.01 (cancer versus normal), 1.89, *p* < 0.05 (cancer versus precancer), and 14.70, *p* < 0.01 (precancer versus healthy control) [28]. Similarly, Chen et al. reported the role of the CDK4 substrate of p16, which is produced by the cyclin-dependent kinase 2 (CDKN2) gene, in oral carcinogenesis, as demonstrated by the aberrant expression of the protein in leukoplakia and OSCC [29]. In this study, no statistically significant variations were found in the expression of GNB2 and TCF7L. This could be attributed to the fact that previous studies have shown that these genes are related to addiction to alcohol and tobacco in patients with systemic conditions, whereas, in the present study, the sample size was low and the patients were otherwise systemically healthy and drug dependence was not severe [17,18].

The current study is the first to perform gene profiling using used toothbrushes. Similar studies have used toothbrushes for person and gender identification, whereas gene profiling has not yet been performed [21,30,31,32,33,34]. The methodology for DNA isolation has been standardized, and this study provides a reference for the future exploration of toothbrushes not only as a forensic tool but also for the early diagnosis of diseases. The advantages of using a toothbrush is that it is noninvasive and the RNA yield is adequate. However, a disadvantage is that the accidental exchange of toothbrushes with family members could alter the results.

Some of the study limitations include the small sample size, duration of toothbrush use, and RNA yield, which were not assessed. Future studies can be conducted to assess the use of toothbrushes in the early detection of oral squamous cell carcinoma.

## 5. Conclusions

Within the limitations of the study, it can be concluded that used toothbrushes are an excellent source of RNA, enabling gene expression analysis for the diagnosis of oral squamous cell carcinoma at an early stage among tobacco and alcohol users. Further studies investigating the ideal protocol for RNA extraction from commercially available toothbrushes are required in order to translate the findings of the present study to routine clinical practice. In addition, variables including the number of bristles, hardness of the bristles, and the use of toothpaste (a potential PCR inhibitor) must be assessed to provide further insight into their influence on the extracted RNA quality and quantity.

## Figures and Tables

**Figure 1 ijerph-19-08052-f001:**
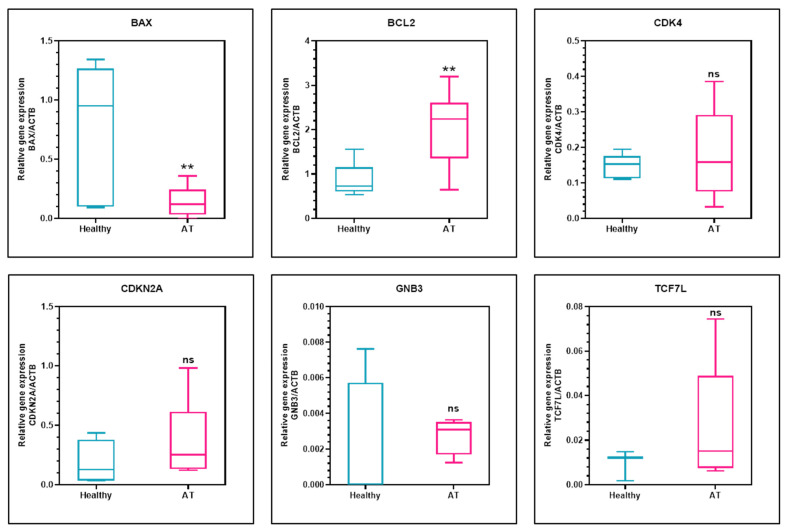
Gene expression of BAX, BCL2, CDK4, CDKN2A, GNB3, and TCF7L2 normalized to ACTB in healthy individuals vs. alcohol and tobacco users (AT). ** *p* ≤ 0.01.

**Table 1 ijerph-19-08052-t001:** List of primers.

Gene	Forward Primer	Reverse Primer	Accession No.
BAX	5′-TCA GGA TGC GTC CAC CAA GAA G-3′	5′-TGT GTC CAC GGC GGC AAT CAT C-3′	NM_001291428
CDK4	5′-CCA TCA GCA CAG TTC GTG AGG T-3′	5′-TCA GTT CGG GAT GTG GCA CAG A-3′	BC005864
BCL2	5′-ATC GCC CTG TGG ATG ACT GAG T-3′	5′-GCC AGG AGA AAT CAA ACA GAG GC-3′	NM_000657.3
CDKN2A	5′-CTC GTG CTG ATG CTA CTG AGG A-3′	5′-GGT CGG CGC AGT TGG GCT CC-3′	NM_001195132
GNB3	5′-GAG TCG GAC ATC AAC GCC ATC T-3′	5′-ATG CCG CAG ATG ATG CTC TCG T-3′	NM_002075.2
TCF7L2	5′-GAA TCG TCC CAG AGT GAT GTC G-3′	5′-TGC ACT CAG CTA CGA CCT TTG C-3′	NM_001198525
ACTB	5′-AGA GCT ACG AGC TGC CTG AC-3′	5′-AGC ATT TCT TCC CGG CCT TT-3′	BC009636

**Table 2 ijerph-19-08052-t002:** PCR cycling conditions.

Stage	Time (min:s)	Temperature (°C)	Cycles
Initial denaturation	10:00	95 °C	1×
Denaturation	2:00	95 °C	
Annealing	0:30	58 °C	40×
Extension	1:00	72 °C	
Melt curve	Increment of 00:05	95–60 °C	1×

**Table 3 ijerph-19-08052-t003:** Relative gene expression (reference gene ACTB).

Marker	Healthy	AT	*p*-Value (Healthy vs. AT)
BAX	0.73645 ± 0.53477	0.13867 ± 0.12014	0.0054 **
BCL2	0.84895 ± 0.36261	1.91001 ± 0.90425	0.0055 **
CDK4	0.14593 ± 0.0309	0.18325 ± 0.11383	0.5144
CDKN2A	0.18127 ± 0.16225	0.3767 ± 0.28963	0.2901
GNB3	0.00191 ± 0.0033	0.00277 ± 0.00091	0.6767
TCF7L2	0.00962 ± 0.00558	0.02558 ± 0.0251	0.3832

AT (alcohol and tobacco users). ** *p* ≤ 0.01.

## Data Availability

Not applicable.

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
