# Peer review of "Role of Toothbrushes as Gene Expression Profiling Tool for Oral Cancer Screening in Tobacco and Alcohol Users"

_ijerph, 2022, doi:10.3390/ijerph19138052_

Round 1

Reviewer 1 Report

The development of tools for early detection of oral cancer is very important. Early detection of oral cancer using toothbrush is an non-invasive method and this research is very interesting. However, I think there are some considerations for this manuscript to be accepted.

Line 63: There are double-space. Please revise.

Line 81: "immune surveillanceDespite..." Please revise.

Line 106: Is group 2 healthy (non oral cancer patients)? I think smoking or alcohol is a "risk factor" for oral cancer, and is not the cause. Examine the relationship between the grade of oral cancer and gene expression in patients with oral cancer will make it more strength as a tool for early detection of cancer. Please consider.

Line 110: "previous history of smoking a minimum of hundred cigarettes in the last five years" Is it true? Isn't there too few 20 cigarettes per year? Please check.

Line 112: Presence of habits such as use of tobacco and alcohol is exclusion criteria. Is it true?

Table 1: Is GNB2 true? GNB3?

Table 1: Is TCF7L true? TCF7L2?

Line 157: p=0.0055**? (double asterisk?)

Line 158: CDKN2A (p=0.2901 instead of 0.5144)?

Line 159-160: Please revise non-bold.

Line 159: GNB3 and TCF7L2?

Table 3: GNB3 and TCF7L2?

Table 3: Please write the full name of AT (Alcohol and Tobacco users) in footnote.

Table 3: p value of BAX is 0.0054** (double asterisk).

Table 3: p value of BCL2 is 0.0055** (double asterisk).

Figure 1: GNB3?

Line 235: Informed consent statement: "Not applicable" is true? This is a clinical study, so informed consent should be obtained from all subjects involved in this study. Please check. 

Author Response

Reviewer 1:

The development of tools for early detection of oral cancer is very important. Early detection of oral cancer using toothbrush is an non-invasive method and this research is very interesting. However, I think there are some considerations for this manuscript to be accepted.

Line 63: There are double-space. Please revise.

Response: Corrected

The disease is multifactorial in etiology with complex pathogenesis right from prolonged exposure to carcinogen. Although several etiologic factors are known to cause the disease, tobacco and alcohol accounts for forty percent of cancers and sixty percent fatality [2].

Line 81: "immune surveillance Despite..." Please revise.

Response: Corrected

Considering pathogenesis following prolonged exposure of oral mucosa to these carcinogens several complex genetic and molecular mechanisms occur that leads to increased mitosis, evasion of apoptosis, and immune surveillance. Despite several recent advances in management of the disease, prognosis and five year survival rate has not seen much improvement.

Line 106: Is group 2 healthy (non oral cancer patients)? I think smoking or alcohol is a “risk factor” for oral cancer, and is not the cause. Examine the relationship between the grade of oral cancer and gene expression in patients with oral cancer will make it more strength as a tool for early detection of cancer. Please consider.

Response: Group 2 patients are tobacco/alcohol consumers and were considered as the study group with a risk for oral cancer. Tobacco/alcohol were stated in the manuscript as a risk factor for oral cancer as advised by the reviewer.

Line 110: "previous history of smoking a minimum of hundred cigarettes in the last five years" Is it true? Isn't there too few 20 cigarettes per year? Please check.

Response: The intention of the study was to establish the potential molecular changes on exposure to tobacco/alcohol with the lowest threshold point. Thus, as in the manuscript there were changes noted in the study group (with minimum of 20 cigarettes per year), it is likely that these changes will only increase in frequency with increased consumption of tobacco/alcohol. This was the scientific rationale behind the setting the sample characteristic with a minimal consumption of 20 cigarettes per year

Line 112: Presence of habits such as use of tobacco and alcohol is exclusion criteria. Is it true?

Response: Corrected

Exclusion criteria

Group I: Presence of other systemic diseases, complications, habits such as use of tobacco and alcohol, pregnancy, lactation.

Group II: Presence of other systemic diseases, complications, pregnancy, lactation

Table 1: Is GNB2 true? GNB3?

Response: Corrected

GNB3

5’-GAG TCG GAC ATC AAC GCC ATC T-3’

5’-ATG CCG CAG ATG ATG CTC TCG T-3’

Table 1: Is TCF7L true? TCF7L2?

Response: Corrected

TCF7L2

5’-GAA TCG TCC CAG AGT GAT GTC G-3’

5’-TGC ACT CAG CTA CGA CCT TTG C-3’

Line 157: p=0.0055**? (double asterisk?)

Response: As advised, the p value less than 0.001 are represented with double asterisk.

Line 158: CDKN2A (p=0.2901 instead of 0.5144)?

Response: As advised, the p value is corrected to p=0.2901

Line 159-160: Please revise non-bold.

Response: Done

The expression of GNB3(p= 0.6767) and TCF7L2 (p=0.3832) showed no statistically significant difference in both the groups. (Table 3).

Line 159: GNB3 and TCF7L2?

Response: The expression of GNB3(p= 0.6767) and TCF7L2 (p=0.3832) showed no statistically significant difference in both the groups. (Table 3).

Table 3: GNB3 and TCF7L2?

Response: Done

GNB3

0.00191±0.0033

0.00277±0.00091

0.6767

TCF7L2

0.00962±0.00558

0.02558±0.0251

0.3832

Table 3: Please write the full name of AT (Alcohol and Tobacco users) in footnote.

Response: As advised, full form of AT is added in the foot note of the table.

Table 3: p value of BAX is 0.0054** (double asterisk).

Response: As advised, double asterisk is added to p value of BAX

Table 3: p value of BCL2 is 0.0055** (double asterisk).

Response: As advised, double asterisk is added to p value of BCL2

Figure 1: GNB3?

Response: Expression of BAX, BCL2, CDK4, CDKN2A, GNB3 and TCFL normalized to ACTB gene in Healthy and Alcohol Tobacco users (AT).

Line 235: Informed consent statement: "Not applicable" is true? This is a clinical study, so informed consent should be obtained from all subjects involved in this study. Please check. 

Response: Informed consent: Consent was obtained from all the patients participating in the study. The same is mentioned in the revised article

Reviewer 2 Report

In this manuscript, several concerns need to be addressed and much important information is missed as follows:

1.      The description of individuals included in the study is missed like the age, sex, …etc.

2.      The authors mentioned in line 15 that “with ~0.5 gm of toothpaste on it” is the type of toothpaste used fixed for all individuals? What is the type of this toothpaste? Also, “gm” should be “g”

3.      The references of the methods used are missed like the method of RNA extraction, the 2–ΔΔCt technique, ….etc.

4.      The table of the primers of analyzed genes should be completed with the accession no. of analyzed genes in GenBank or the references.

5.      In all tables’ footnotes: The full term of all abbreviations used within the table should be clarified in the footnote.

6.      In table 3: a huge number of significant numbers is presented. E.g. 0.73645±0.53477 should be 0.74±0.54. Also, 0.0054 should be 0.01.

7.      Conclusion is very concise and needs to be supported with further perspectives.

8.      Many formatting errors exist. E.g. line 81, surveillanceDespite, line 113, alcohol, Pregnancy,…etc.

Author Response

Reviewer 2:

In this manuscript, several concerns need to be addressed and much important information is missed as follows:

  1. The description of individuals included in the study is missed like the age, sex, …etc.

Response: Included the male and female count and the age range.

  1. The authors mentioned in line 15 that “with ~0.5 gm of toothpaste on it” is the type of toothpaste used fixed for all individuals? What is the type of this toothpaste? Also, “gm” should be “g”

Response: Mentioned the toothpaste name and use details. Changed “gm” to “g”.

  1. The references of the methods used are missed like the method of RNA extraction, the 2–ΔΔCt technique, ….etc.

Response: Cited the references used for RT-qPCR protocol.

  1. The table of the primers of analyzed genes should be completed with the accession no. of analyzed genes in GenBank or the references.

Response: Included accession numbers for all the primers in the table 1.

  1. In all tables’ footnotes: The full term of all abbreviations used within the table should be clarified in the footnote.

Response:  As advised, full forms are mentioned in the table foot notes

  1. In table 3: a huge number of significant numbers is presented. E.g. 0.73645±0.53477 should be 0.74±0.54. Also, 0.0054 should be 0.01.

Response: Since the result are already in decimal points, the authors feel by rounding off we could potentially increase/decrease the value dramatically.

  1. Conclusion is very concise and needs to be supported with further perspectives.

Response: As advised, future perspective is added to the conclusion.

  1.       Many formatting errors exist. E.g. line 81, surveillanceDespite, line 113, alcohol, 

pregnancy,…etc.

Response: Done

Round 2

Reviewer 1 Report

I have no more comments.

Reviewer 2 Report

no further comments to be addressed